

# A Flexible Algorithm for Network Design Based on Information Theory

Rona L. Thompson[1] and Ignacio Pisso[1]

[1]NILU – Norsk Institut for Luftforskning

*Correspondence to*: Rona L. Thompson (rlt@nilu.no)

**Abstract.** A novel method for atmospheric network design is presented, which is based on Information Theory. The method does not require calculation of the posterior uncertainty (or uncertainty reduction) and, therefore, is computationally more efficient than methods that require this. The algorithm is demonstrated in two examples, the first looks at designing a network for monitoring $CH_4$ sources using observations of the stable carbon isotope ratio in $CH_4$ ($\delta^{13}C$), and the second looks at designing a network for monitoring fossil fuel emissions of $CO_2$ using observations of the radiocarbon isotope ratio in $CO_2$

($\Delta^{14}CO_2$).

## 1. Introduction

The optimal design of any observing network is an important problem in order to maximise the information obtained with minimal cost. In atmospheric sciences, observing networks include those for weather prediction as well as for air quality and the monitoring of greenhouse gases (GHGs). For air quality and GHGs, one essential purpose of the observation network is to

learn about the underlying sources and, where relevant, the sinks. This application is based on inverse methodology in which knowledge about some unknown variables, in this case the sources (and sinks), can be determined by indirect observations, that is the atmospheric concentrations or mixing ratios, if there is a model or function that relates the unknown variables to the observations. Inverse methodology provides a means to relate the observations to the unknown variables and provides an optimal estimate of these (Tarantola, 2005).


In atmospheric sciences, the methodology is most often derived from Bayes' Theorem, which describes the conditional probability of the state variables, x, given the observations, y:

$$P(x|y) = \frac{P(y|x)P(x)}{P(y)} \qquad (1)$$

Assuming a Gaussian probability density function (pdf), the following cost function can be derived (Rodgers, 2000):

$$J(\mathbf{x}) = \frac{1}{2}(\mathbf{x} - \mathbf{x_b})^T \mathbf{B}^{-1}(\mathbf{x} - \mathbf{x_b}) + \frac{1}{2}(H(\mathbf{x}) - \mathbf{y})^T \mathbf{R}^{-1}(H(\mathbf{x}) - \mathbf{y}) \qquad (2)$$

The **x** for which $J(\mathbf{x})$ is minimum is the state vector that minimizes the sum of two distances: one in the observation space, between the modelled, $H(\mathbf{x})$, and observed, **y**, variables, and the other in the state space, between **x** and a prior estimate of state variables, $\mathbf{x_b}$. These two distances are weighted by the matrices **R** and **B**, which are respectively, the observation error



covariance and prior error covariance. Expressions for the centre and variance of the posterior pdf of **x** are given by e.g.
Tarantola, 2005.

The choice of the locations for the observations has important consequences for how well the state variables can be constrained. Increasing the number of observations will decrease the dependence of the solution on $\mathbf{x_b}$, but where those observations are made is also a critical consideration and depends how they relate to the state variables, as described by the transport operator,
$H(\mathbf{x})$. Here only the linear transport case is considered in which this operator can be defined as the matrix **H**.

In practical applications of network design, there is usually a predefined budget that would allow the establishment of a given number of sites, either to create a new network or to add to an existing one. The possible locations of sites is usually a predefined set since these need to fulfil certain criteria, e.g., access to the electrical grid, internet connection, road access, an
existing building on site to house instruments, the agreement of the property owner, and may include having an existing tower if measurements are to be made above the surface layer. Thus, the question is often: which potential sites should be chosen to provide the most information about the sources and sinks?

There are already a number of examples of network design in the framework of atmospheric monitoring in the scientific
literature. An early example is the optimization of a global network for $CO_2$ observations to improve knowledge of the terrestrial $CO_2$ fluxes (Gloor et al 2000; Patra and Maksyutov 2002; Rayner et al 1996). These studies dealt only with small dimensional problems, i.e., with few state variables and relatively low frequency observations and, thus, small **B** and **H** matrices, and the criteria by which the network was chosen was minimizing the posterior uncertainty. Gloor et al. solved the problem using a Monte Carlo method (specifically Simulated Annealing) but they found this method took considerable time
to converge and up to $5 \times 10^5$ iterations were needed. Patra and Makysutov used a less computationally demanding approach, the Incremental Optimization method, which is based on the "divide and conquer" algorithm principle. In this method, the problem to solve is broken down into steps, i.e., sequentially choosing the best site from the set of potential sites and correspondingly depleting this set by one with each step. In the Incremental Optimization approach only $\sum_{i=1}^{k}(p - i + 1)$ calculations are needed, where $k$ is the number of sites to select and $p$ the number of potential sites to choose from. The
Incremental Optimization approach, however, may lead to a different selection of sites compared to testing all possible combinations of sites, which would involve $p!/(k!(p - k)!)$ calculations, but this in many cases may be a prohibitively large number.

More recently, the problem of network design has been addressed in the context of regional networks for GHG observations
(Lucas et al., 2015; Nickless et al., 2015). Again, in both these studies the metric for selecting the network was the posterior uncertainty, either by using the trace of the posterior error covariance matrix, which is equivalent to minimizing the mean square uncertainty for all grid cells (Lucas et al., 2015) or by minimising the sum of the posterior error covariance matrix,





which accounts also for the covariance of uncertainty between grid cells (Nickless et al., 2015). These studies both used Monte Carlo approaches (specifically, Genetic Algorithms) to find the network minimizing the selected metric.


However, for large problems any metric involving the posterior uncertainty becomes a bottleneck, if not unworkable, since the calculation of the posterior error covariance matrix, $\mathbf{A}$ requires inverting the matrix $\mathbf{H}^{\mathrm{T}}\mathbf{R}^{-1}\mathbf{H} + \mathbf{B}^{-1}$ which has dimensions of $n \times n$ where $n$ is the number of state variables. For this reason, methods were proposed based on criteria considering how well a network resolves the atmospheric variability or "signal" or, in other words, how well they sample regions of significant

heterogeneity (Shiga et al., 2013). In this approach, the atmospheric signal (e.g. mixing ratio) is modelled using an atmospheric transport model and a prior flux estimate and sites are sequentially added to the network so that the distance of any grid cell from an observation site is within some pre-determined correlation scale length. For this method, the number of calculation steps is equal to the sites to be selected (Shiga et al., 2013). Although computationally very efficient, this method does not consider the information gained about the state variables but only the optimal sampling of atmospheric variability.


An alternative method, but also based on the consideration of atmospheric variability, is to consider how "similar" the atmospheric signal is between potential sites in a network and to reduce the number sites leaving only those with significantly different signals (Risch et al., 2014). Risch et al. applied a clustering method to cluster sites with similar signals (i.e., strongly correlated sites) and individual sites were removed from each cluster based on the premise that they did not contribute any

significant new information, whereas sites in clusters of one member were all retained. However, as in the method of Shiga et al. (2013), this approach does not consider the information gained about the state variables and how atmospheric transport alone may influence the variability at each site.

Here a method for network design is proposed based on Information Theory. This method requires precomputed transport

operators for each potential site, so-called site "footprints" or "source receptor relationships (SRRs)", which can be calculated directly using a Lagrangian atmospheric transport model (Seibert and Frank, 2004) or from forward calculations of a Eulerian transport model for each source (Rayner et al., 1999; Enting, 2002). The method can be applied to the problem of creating a new network or expanding an existing one, and can be applied to observations of mixing ratios, isotopic ratios, column measurements, or a combination of these. It provides an alternative criterion to the posterior uncertainty (or uncertainty

reduction) to assess a potential network and can be used with either Incremental Optimization or Monte Carlo approaches. It has a number of advantages compared to previous methods: i) it does not require the inversion of any large matrix (e.g., to the calculate posterior uncertainty) making it computationally efficient, ii) it accounts for spatial correlations in the state variables, and iii) it allows for an exact formulation of the problem to be solved, i.e., what is the improvement in knowledge about the unknown variables. On the other hand, it requires linearity of the operator from the state space to the observation space, which

is not the case for methods examining only atmospheric variability.



Two example applications are presented, which are based on real-life network design problems. The first considers adding measurements of the stable isotope ratio of $CH_4$, i.e., $\delta^{13}C$ to a subset of existing sites measuring $CH_4$ mixing ratios in order to maximise the information about $CH_4$ sources. The second considers designing a network for $\Delta^{14}CO_2$ measurements to maximise the information about fossil fuel emissions of $CO_2$.

## 2. Methodology

In Information Theory, the information content of a measurement can be thought of as the amount by which knowledge of some variable is improved by making the measurement, and the entropy is the level of information contained in the measurement (Rodgers, 2000). In this case, one can consider the pdf as a measure of knowledge about the state variables and the information provided by a measurement can be found by comparing the entropy of the pdfs before and after measurement was made. Furthermore, the information content of the measurement is equal to the reduction in entropy. In the application of network design, all observations within the potential network are considered as one "measurement".

The entropy, $S$ of the pdf given by $P(x)$ is:

$$S\big(P(x)\big) = - \int P(x)\ln\big(P(x)\big) \tag{3}$$

And the information content, $I$ is the reduction in entropy after a measurement is made:

$$I = S\big(P(x)\big) - S\big(P((x|y))\big) \tag{4}$$

Where $P(x)$ is the prior pdf (before measurement) and $P(x|y)$ is posterior pdf (after the measurement, $y$). The entropy is given by integrating Eq. 3 over the bounds $-\infty$ to $+\infty$ (Rodgers, 2000) which for a Gaussian pdf of a scalar variable is:

$$S = \ln\left(\sigma(2\pi e)^{\frac{1}{2}}\right) \tag{5}$$

where $\sigma$ is the standard deviation. In the multivariate case with $m$ variables the entropy is given by:

$$S = \sum_{i=1}^{m} \ln(2\pi e \lambda_i)^{\frac{1}{2}} \tag{6}$$

where $\lambda_i$ is an Eigenvalue of the error covariance matrix. By rearrangement one can write:

$$S = \sum_{i=1}^{m}\left(\ln(2\pi e)^{\frac{1}{2}} + \ln\lambda_i^{\frac{1}{2}}\right) \tag{7}$$

$$S = m\ln(2\pi e)^{\frac{1}{2}} + \frac{1}{2}\ln(\textstyle\prod \lambda_i) \tag{8}$$

$$S = m\ln(2\pi e)^{\frac{1}{2}} + \frac{1}{2}\ln|\mathbf{B}| \tag{9}$$

In Eq. 9 $|\mathbf{B}|$ is the determinant of the prior error covariance matrix using the fact that the determinant of a symmetric matrix is equal to the product of its eigenvalues. Similarly, the entropy for the posterior pdf can be derived, giving the information content as:

$$I = \frac{1}{2}\ln|\mathbf{B}| - \frac{1}{2}\ln|\mathbf{A}| \tag{10}$$



Where **A** is the posterior error covariance matrix. In this case the determinant can be thought of defining the volume in state space occupied by the pdf, which describes the knowledge about the state, thus $I$ is the change in the log of the volume when observation is made. From Eq. 10 one can derive:

$$I = \frac{1}{2}\ln|\mathbf{B}\mathbf{A}^{-1}| \tag{11}$$

And given that the inverse of **A** is equal to the Hessian matrix of $J(\mathbf{x})$ (Eq. 2),

$$\mathbf{A}^{-1} = \mathbf{H}^{\mathrm{T}}\mathbf{R}^{-1}\mathbf{H} + \mathbf{B}^{-1} \tag{12}$$

one obtains

$$I = \frac{1}{2}\ln|\mathbf{B}\mathbf{H}^{\mathrm{T}}\mathbf{R}^{-1}\mathbf{H} + \mathbf{I}| \tag{13}$$

where **R** is the observation error covariance matrix, **H** is the model operator (for atmospheric observations it is the atmospheric
transport operator) and **I** is the identity matrix.

The principle of this network design method is to choose the sites that maximise the information, and this criterion can be used in either the Incremental Optimization or Monte Carlo approach. The Incremental Optimization approach is computationally efficient, requiring only $\sum_{i=1}^{k}(p - i + 1)$ calculations and delivers, if not the same, at least similar results to testing all possible
combinations of sites (Patra and Maksyutov, 2002).

The calculation of the matrix $\mathbf{B}\mathbf{H}^{\mathrm{T}}\mathbf{R}^{-1}\mathbf{H} + \mathbf{I}$ can be quite fast since **H** and **R** can be made quite small. **H** does not need to represent all observations for each site, but only the average observation corresponding to different levels of uncertainty or "characteristic observations". In the case that observations at each site have only one characteristic uncertainty, then **H** will
have dimension $k \times n$ where $n$ is the number of state variables, and **R** will be $k \times k$, and in practice **R** is most often diagonal. In the case that the uncertainty of an observation at a given site varies depending on when it was made, e.g., daytime or nighttime, then the dimension of **H** will be $2k \times n$. The computationally demanding step is the calculation of the matrix determinant. However, this calculation can be made very efficient if the matrix $\mathbf{B}\mathbf{H}^{\mathrm{T}}\mathbf{R}^{-1}\mathbf{H} + \mathbf{I}$ is decomposed into **B** and $(\mathbf{H}^{\mathrm{T}}\mathbf{R}^{-1}\mathbf{H} + \mathbf{B}^{-1})$, which are both symmetric positive definite matrices, and using the fact that the log of the determinant of a
symmetric positive definite matrix can be calculated as the trace of the log of the lower triangular matrix of the Cholesky decomposition:

$$I = \frac{1}{2}\ln(|\mathbf{B}|) + \frac{1}{2}\ln(|\mathbf{H}^{\mathrm{T}}\mathbf{R}^{-1}\mathbf{H} + \mathbf{B}^{-1}|) \tag{14}$$

$$= \mathrm{tr}\big(\ln(\mathbf{L})\big) + \mathrm{tr}\big(\ln(\mathbf{M})\big) \tag{15}$$

where $\mathbf{B} = \mathbf{L}\mathbf{L}^{\mathrm{T}}$ and $\mathbf{H}^{\mathrm{T}}\mathbf{R}^{-1}\mathbf{H} + \mathbf{B}^{-1} = \mathbf{M}\mathbf{M}^{\mathrm{T}}$ where **L** and **M** are the lower triangular matrices. Note, that if temporal correlations
in **B** can be ignored, then **B** needs only to be formulated for a single time step, i.e., $\mathbf{B}_t$, which is a considerably smaller matrix than **B**, and $\mathbf{H}^{\mathrm{T}}\mathbf{R}^{-1}\mathbf{H} + \mathbf{B}^{-1}$ can be calculated stepwise adding $\mathbf{B}_t^{-1}$ for each timestep. Furthermore, $\mathbf{B}_t^{-1}$ (or $\mathbf{B}^{-1}$) only needs to be calculated once since it does not change with choice of sites. In this case the information content is simply:

$$I = q\,\mathrm{tr}(\ln(\mathbf{L})) + \mathrm{tr}(\ln(\mathbf{M})) \tag{16}$$



Where $q$ is the number of timesteps and $\mathbf{L}$ in this case is the lower triangular matrix of $\mathbf{B}_t$.

## 3. Examples

### 3.1. Enhancing a network for estimating sources of CH₄

This example considers the enhancement of a network of atmospheric measurements of $CH_4$ mixing ratios by adding observations of stable isotopic ratios, $\delta^{13}C$ at a selected number of sites within the existing network in order to improve knowledge of the different $CH_4$ sources. For the example, the case of the Integrated Carbon Observing System (ICOS) network (https://www.icos-cp.eu) in Europe is used, which consists of 24 operational sites in geographical Europe measuring $CH_4$ mixing ratios (Table 1). In this hypothetical case, the budget is available to equip 5 of the 24 sites with in-situ instruments measuring $\delta^{13}C$ at hourly frequency, as is now possible with modern instrumentation (Menoud et al., 2020). The problem can thus be formulated as: given the existing information provided by 24 sites measuring $CH_4$ mixing ratio, which sites are the best to choose for the additional $\delta^{13}C$ observations?

The $\delta^{13}C$ value is the ratio of $^{13}C$ to $^{12}C$ in a sample relative to a reference value measured in per mil (‰):

$$\delta^{13}C = \left( \frac{R_{sam}}{R_{ref}} - 1 \right) \times 1000 \qquad (17)$$

The $\delta^{13}C$ value in the atmosphere varies as a result of variations in the $\delta^{13}C$ value of the sources, the oxidation of $CH_4$ in the atmosphere and in soils, and atmospheric transport. Sources of $CH_4$ can be grouped according to their characteristic $\delta^{13}C$ value, with microbial sources being the most depleted in $^{13}C$, while thermogenic sources such as from oil, gas, and coal, are less depleted, and pyrogenic sources, such as biomass burning, are the least depleted (Fisher et al., 2011; Dlugokencky et al., 2011; Brownlow et al., 2017). In this example, $CH_4$ sources were grouped into anthropogenic microbial sources, namely, agriculture and waste (agw), fossil sources, namely fossil fuel and geological emissions (fos), biomass burning sources (bbg), natural microbial sources, principally wetlands (wet) and the ocean source (oce). The change in $CH_4$ mixing ratio from all sources can thus be written as:

$$\Delta c = \mathbf{H}\mathbf{x}_{agw} + \mathbf{H}\mathbf{x}_{wet} + \mathbf{H}\mathbf{x}_{fos} + \mathbf{H}\mathbf{x}_{bbg} + \mathbf{H}\mathbf{x}_{oce} \qquad (18)$$

Where $\Delta c$ is the change in $CH_4$ mixing ratio, $\mathbf{x}$ is the vector of fluxes, $\mathbf{H}$ is the transport operator. Analogously, the change in $\delta^{13}C$ can be defined as:

$$\Delta \delta^{13}c = \mathbf{H}\delta_{agw}\mathbf{x}_{agw} + \mathbf{H}\delta_{wet}\mathbf{x}_{wet} + \mathbf{H}\delta_{fos}\mathbf{x}_{fos} + \mathbf{H}\delta_{bbg}\mathbf{x}_{bbg} + \mathbf{H}\delta_{oce}\mathbf{x}_{oce} \qquad (19)$$

Where $\delta_x$ is the isotopic signature for each source type. Therefore, the transport operator for an observation of the change in $\delta^{13}C$ is the just the transport operator $\mathbf{H}$ but scaled by $\delta_x$ for each source.

For this example, SRRs were calculated for all 24 sites in the ICOS network using the Lagrangian particle dispersion model, FLEXPART (Pisso et al., 2019) driven with ERA Interim reanalysis wind fields. Retro-plumes were calculated for 10 days



backwards in time from each site at hourly frequency. The SRRs were saved at 0.5°×0.5° resolution over the European domain of 12°W to 32°E and 35°N to 72°N and averaged over all observations within a month to give a monthly mean SRR for each site.

The uncertainty in the $\delta^{13}$C measurements was set to the same value for each site, that is, at 0.07‰ based on experimental

values (Menoud et al., 2020). Similarly, the uncertainty in $CH_4$ mixing ratio measurements was also set to the same value at all sites, at 5 ppb (WMO, 2009). The prior uncertainty, $\sigma$ for each grid cell was calculated as 0.5 times the prior flux, with a lower threshold equal to the 1 percentile value of all grid cells with non-zero flux for the smallest flux source. The spatial correlation between grid cells was calculated based on exponential decay over distance with a correlation scale length of 250 km over land. The prior error covariance matrix was then calculated as:

$$\mathbf{B} = \Sigma\mathbf{C}\Sigma \qquad (20)$$

Where $\mathbf{C}$ is the spatial correlation matrix and $\Sigma$ is a diagonal matrix with the diagonal terms equal to the prior uncertainties for each grid cell.

For this example, the optimal network was found for three different scenarios: 1) monitoring all sources in EU27 countries

plus UK, Norway, and Switzerland (EU27+3), 2) monitoring only anthropogenic sources in EU27+3, and 3) as in scenario 1 but ignoring the existing information provided by $CH_4$ mixing ratios at all sites.

For these scenarios the influence of the fluxes that are not the target of the network needs to be projected into the observation space and included in the $\mathbf{R}$ matrix. For example, in scenario 1 this is the influence of fluxes outside EU27+3, and in scenario

2 it is the influence of all non-anthropogenic sources plus the influence of fluxes outside EU27+3. This is calculated as:

$$\mathbf{R} = \mathbf{H}\mathbf{B}_{other}\mathbf{H}^T + \mathbf{R}_{meas} \qquad (21)$$

Where $\mathbf{R}_{meas}$ is simply the prior measurement uncertainty and $\mathbf{B}_{other}$ is the prior error covariance matrix for the other (i.e., non-target) fluxes.

For all scenarios the choice of the first four optimal sites was the same, that is, IPR, SAC, KIT, and LIN, while the last site chosen was KRE in scenarios 1 and 2 (Fig. 1), and LUT in scenario 3. All chosen sites are strongly sensitive to anthropogenic emissions, and the choice to optimize all sources or only anthropogenic sources made no difference in this example, likely because the natural sources (predominantly wetlands) are a relatively small contribution to the total $CH_4$ source in Europe (only 12%). On the other hand, ignoring existing information provided by $CH_4$ mixing ratios, led to LUT being chosen over

KRE, likely because LUT provides a stronger constraint on the region with the largest emissions and diverse sources, i.e., Benelux (Fig. 2), which is more important in the absence of $CH_4$ mixing ratio data.

**3.2 Network of $^{14}CO_2$ measurements for fossil fuel emissions**



This example concerns the establishment of a network for measurements of radiocarbon dioxide, $^{14}CO_2$, which can be used as a tracer for fossil fuel $CO_2$ emissions, since fossil fuel contains no $^{14}C$ its combustion depletes the atmospheric background value of $^{14}CO_2$ (Turnbull et al., 2009). Similar to the previous example, the ICOS network is used, which also has $CO_2$ measurements at 24 sites in Europe. The hypothetical problem can be formulated as follows: if there is budget to equip 10 sites in the ICOS network with weekly flask samples for $^{14}CO_2$ analysis, which sites should be chosen to gain the most knowledge of fossil fuel emissions? In this case, only weekly measurement frequency is examined as $^{14}CO_2$ measurements cannot be made continuously and the measurement method, either via counting radioactive decay or by accelerator mass spectrometry, is costly and time consuming. The optimization problem needs to consider the information already brought by the $CO_2$ measurements at all sites (in this example hourly measurements) and, in addition, the influence on the atmospheric signal from other sources, which may change the sensitivity of a site to fossil fuel emissions.

Measurements of $^{14}CO_2$ are reported as the ratio of $^{14}CO_2$ to $CO_2$ relative to a reference ratio and given in units of per mil (‰):

$$\Delta^{14}C = \left(\frac{R_{sam}}{R_{ref}} - 1\right) \times 1000 \tag{22}$$

Since fossil fuels contain no $^{14}C$, its isotopic ratio is -1000‰. Other than fossil fuels, atmospheric values of $\Delta^{14}CO_2$ are determined by natural production of $^{14}CO_2$ in the stratosphere, nuclear power and spent fuel processing plants, and from ocean and land biosphere fluxes, as well atmospheric transport. Ocean fluxes affect $^{14}CO_2$ since the ocean is not in isotopic equilibrium with the atmosphere owing to higher values of atmospheric $^{14}CO_2$ in the past due to nuclear bomb testing, and similarly for plant respiration fluxes of $CO_2$ (Bozhinova et al., 2014).

The change $CO_2$ mixing ratio can be described as follows:

$$\Delta c = \mathbf{H}\mathbf{x}_{fos} + \mathbf{H}\mathbf{x}_{pho} + \mathbf{H}\mathbf{x}_{res} + \mathbf{H}\mathbf{x}_{oce} \tag{23}$$

Where $x_{fos}$ is the fossil fuel emission, $x_{pho}$ is the land biosphere photosynthesis flux, $x_{res}$ is the land biosphere respiration flux, and $x_{oce}$ the net ocean flux. A similar expression for the change in $\Delta^{14}CO_2$ can be derived following (Bozhinova et al., 2014) as:

$$\Delta^{14}c = \mathbf{H}\Delta_{fos}\mathbf{x}_{fos} + \mathbf{H}\Delta_{pho}\mathbf{x}_{pho} + \mathbf{H}\Delta_{res}\mathbf{x}_{res} + \mathbf{H}\Delta_{oce}\mathbf{x}_{oce} + \mathbf{H}\Delta_{nuc}\mathbf{x}_{nuc} \tag{24}$$

Where $\Delta^{14}c$ is the change in $\Delta^{14}CO_2$ and $\Delta_x$ is the isotopic signature of the corresponding source and the term $\mathbf{H}\Delta_{nuc}\mathbf{x}_{nuc}$ is the production of $^{14}CO_2$ from nuclear facilities. There is a term missing from Eq. 23 and 24, namely the stratospheric production of $CO_2$ and $^{14}CO_2$. This term is ignored as the direct stratospheric contribution is negligible for the time and space domain considered by the Lagrangian model since the observations are close to the surface. Equation 24 can be further simplified by removing the term $\mathbf{H}\Delta_{pho}\mathbf{x}_{pho}$, since photosynthesis, although affects the $^{14}CO_2$ mixing ratio does not affect $\Delta^{14}CO_2$ (Turnbull et al., 2009). Furthermore, the ocean and respiration fluxes can be split into a background term and a disequilibrium term, $\Delta_{bg} + \Delta_{ocedis}$ and $\Delta_{bg} + \Delta_{resdis}$, respectively. As for photosynthesis, the background terms for ocean and respiration fluxes do not





change $\Delta^{14}CO_2$, but only the disequilibrium terms. For the domain in consideration, these terms are much smaller than that of fossil fuels and are ignored as in (Bozhinova et al., 2014). With these simplifications, Eq. 24 becomes:

$$\Delta^{14}c = \mathbf{H}\Delta_{fos}\mathbf{x}_{fos} + \mathbf{H}\Delta_{nuc}\mathbf{x}_{nuc} \qquad (25)$$

Since $x_{nuc}$ is pure $^{14}CO_2$, $\Delta_{nuc}$ would be infinite, therefore, the approach of (Bozhinova et al., 2014) is used and $\Delta_{nuc}$ is approximated as the ratio of the activity of the sample and the referenced standard giving $\Delta_{nuc} \approx 0.7 \times 10^{15}$ ‰.


Because, in this example, only the fossil fuel emissions are the unknown variables and the target of the network, the matrix $\mathbf{B}$ corresponds only to the uncertainty in the fossil fuel emissions and is resolved monthly. The other terms influencing $CO_2$ and $\Delta^{14}CO_2$ are projected into the observation space and included in the $\mathbf{R}$ matrix using Eq. 21. For the $\Delta^{14}CO_2$ observations, $\mathbf{B}_{other}$ is only the nuclear source, and for $CO_2$ observations, $\mathbf{B}_{other}$ includes photosynthesis and respiration, the sum of which is Net

Ecosystem Exchange (NEE) and the ocean flux, for which the effect on the observed $CO_2$ signal is very small and is thus ignored here. For both NEE and nuclear emissions, an uncertainty of 0.5 times the value in each grid cell was used to calculate $\mathbf{B}_{other}$ with a spatial correlation length of 250 km. Since NEE fluxes have large diurnal and seasonal cycles which co-vary with atmospheric transport, for the $CO_2$ observations, $\mathbf{R}$ was calculated using $\mathbf{H}$ and $\mathbf{B}$ resolved for day and night, and monthly. Note, only one uncertainty value was calculated for each site, which represents the annual mean uncertainty for a daytime

observation. Each site has a different uncertainty for $CO_2$ mixing ratio and $\Delta^{14}CO_2$ depending on the influence of NEE fluxes and nuclear emissions, respectively. This can be simply interpreted in terms of a signal to noise ratio. For example, for $CO_2$ mixing ratios where there is a large influence of NEE the timeseries becomes noisier and similarly for the influence of nuclear emissions on $\Delta^{14}CO_2$ observations. The measurement uncertainty, $\mathbf{R}_{meas}$, was set to the same value for each site, that is, at 2‰ for $\Delta^{14}CO_2$ (Turnbull et al., 2007) and 0.05 ppm for $CO_2$ mixing ratio measurements (WMO, 2018).


For this example, SRRs were calculated for all 24 sites in the ICOS network using FLEXPART with retro-plumes calculated for 5 days backwards in time from each site at hourly frequency. The SRRs were saved at 0.5°×0.5° and 3-hourly resolution over the European domain of 15°W to 35°E and 30°N to 75°N and were averaged to give mean day and night SRRs for each month for each site. Estimates of NEE fluxes were used from the Simple Biosphere Model - Carnegie Ames Stanford Approach

(SiBCASA) and were resolved 3-hourly (Schaefer et al., 2008), estimates of nuclear emissions were used from CHE project (Potier et al., 2022) and were an annual climatology, and estimates of fossil fuel emissions were from GridFED at monthly resolution (Jones et al., 2020).

Figure 3 shows the uncertainty in the observation space at each site due to the influence of uncertainties in NEE and nuclear

emissions on $CO_2$ mixing ratios and $\Delta^{14}CO_2$ values, respectively. For $CO_2$, sites in western Europe have the largest uncertainties, while sites in northern Scandinavia and southern Europe have smaller uncertainties following the pattern of NEE amplitude. For $\Delta^{14}CO_2$, most sites have only small uncertainties owing to nuclear emissions, but two notable exceptions are NOR and KIT, and both are close to large nuclear sources.



The optimal sites in the order selected are: SAC, KIT, LUT, KRE, STE, LIN, GAT, IPR, TRN and TOH (Fig. 4). If the prior

error covariance matrix, **B**, and the transport operator, **H**, are not resolved monthly but only annually, the optimal sites differ

by only one site, namely HPB is chosen instead of TRN. If the existing information provided by $CO_2$ mixing ratios is ignored

(i.e., the network is designed only considering information from $\Delta^{14}CO_2$), then the choice of optimal sites differs slightly and

TRN and TOH are no longer selected but OPE and LMP. The choice of LMP may seem unexpected at first, but it is close to

an emission hotspot in Tunis, Tunisia (Fig. 5). The reason this site is not selected when the information from $CO_2$ mixing ratios

is included is presumably because the $CO_2$ mixing ratio already provides a reasonable constraint on the fossil fuel emissions

with the NEE signal being relatively small.

## 4. Discussion

One question that arises, is how does this method compare to methods based on the analysis of the variability in the timeseries

at the different sites? To answer this question, a clustering method was applied to the example of designing a network for fossil

fuel $CO_2$ emissions. For this, a timeseries of $\Delta^{14}CO_2$ was generated for each of the 24 sites using Eq. 25 (see the supplementary

material for plots of the timeseries). The values were generated hourly but since generally only daytime values are used in

inverse modelling, data were selected for the time interval 12:00 to 15:00. A dissimilarity matrix was calculated for the 24

timeseries' (using the R function proxy::dist with the Dynamic Time Warp (DTW) method (Giorgino, 2009)). The Divisive

Hierarchical Clustering method (R function cluster::diana) was applied to the dissimilarity matrix stopping at 10 clusters. The

first cluster contained 13 sites, that is, those with little signal (e.g. JFJ, CMN, and ZEP). Two clusters contained two sites,

namely, IPR and KRE, and OPE and TRN, while the remaining clusters contained only one site. Based on the principle of

choosing sites that display different signals, one would choose the sites which are in a cluster of one. This would lead to the

choice of GAT, KIT, LIN, LUT, SAC, STE and TOH. These 7 sites are also chosen by the method based on information

content. However, the question is how to choose the remaining 3 sites from clusters with more than one site? For this there is

no single answer. Moreover, the sites that are the most dissimilar are not necessarily those that will provide the most

information about the target fluxes of the network, since the reasons for dissimilarity are various, e.g., having little signal,

being sensitive to sources that are not the target of the network, or owing to distinct atmospheric circulation patterns. While

sites with high degrees of similarity may both offer a strong constraint, and both be valuable to a network (in this example IPR

and KRE were in the same cluster but both sites are chosen in the method based on information content).

In the examples presented, the atmospheric transport matrix, **H**, and the matrix, **B**, were resolved at 0.5°×0.5° (and considered

only land grid cells) and monthly. The size of the matrix **B** (and the matrix $\mathbf{H}^T\mathbf{R}^{-1}\mathbf{H} + \mathbf{B}^{-1}$) for the example on a network for

fossil fuel $CO_2$ emissions was ~11 Gb. However, in the case of finer spatial resolution or a larger domain, which means the

size of the matrices exceeds the available memory, it is still possible to use this method as long as **B** and



$\mathbf{H}^T\mathbf{R}^{-1}\mathbf{H} + \mathbf{B}^{-1}$ defined for one time step do not exceed the memory. In this case, the problem can be solved by summing the information content calculated separately for each time step. Disaggregating the problem in this way does not lead to the same value of information content as when all time steps are considered together, however, the choice of sites is nearly the same; for the example of a network for fossil fuel $CO_2$ emissions the two methods (i.e., disaggregating versus not disaggregating)

differed by only one site. For the example of a fossil fuel network, the total computation time was ~3 hours using multithreaded parallelization on 8 cores.

## 5. Conclusions

A method for designing atmospheric observation networks is presented based on Information Theory. This method can be applied to any type of atmospheric data: mixing ratios, aerosols, isotopic ratios, as well as total column measurements. In

addition, the method allows the network to be designed with or without considering existing information, which may also be of a different type, e.g., mixing ratios of a different species or isotopic ratios. Since the method does not require inverting any large matrices (e.g., for the calculation of posterior uncertainties) and the calculation of $\mathbf{B}^{-1}$ only needs to be performed once, it is very efficient and can be used also on large problems. The only constraint is that the matrices $\mathbf{B}$ and $\mathbf{H}^T\mathbf{R}^{-1}\mathbf{H} + \mathbf{B}^{-1}$ defined for one timestep do not exceed the available memory. The algorithm allows the exact problem to be defined, that is, to target

specific emission sources or regions. Two examples are presented, the first is to select sites from an existing network of $CH_4$ mixing ratios for additional measurements of $\delta^{13}C$ to constrain emissions in EU countries (plus Norway, Switzerland and the UK), and the second to select sites from an existing network of $CO_2$ mixing ratios for additional measurements of $\Delta^{14}CO_2$ to monitor fossil fuel $CO_2$ emissions. These examples demonstrated that the optimal network differs depending on its exact purpose, e.g., is the network targeting emissions over the whole domain or for a specific region, and should existing information

be considered or not, and thus it is important that the method of network design is able to account for these considerations.

### Code availability

The R code for the network design algorithm presented in this paper will be made available from Zenodo.

### Author contributions

R. Thompson developed the algorithm, wrote the code, and carried-out the examples. I. Pisso contributed to the modelling of

$\Delta^{14}CO_2$, discussed the algorithm and provided feedback on the manuscript.

### Competing interests

The authors declare that they have no conflict of interest.



## Acknowledgements

This work was supported by the European Commission, Horizon 2020 Framework Programme (VERIFY, grant no. 776810).
We would like to acknowledge J. Müller for preparing the LPX-Bern simulations of wetland $CH_4$ fluxes and E. Potier and Y.
Wang for providing the estimates of nuclear emissions of $^{14}CO_2$.

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





**Table 1**. List of sites in the ICOS network (only those in geographical Europe are included). The sampling height that was used in this study is shown, which is the highest sampling height at each site.

| Site ID | Full name | Latitude | Longitude | Site Altitude (masl) | Sampling Height (magl) |
|---|---|---|---|---|---|
| CMN | Monte Cimone, Italy | 44.19 | 10.70 | 2165 | 8 |
| GAT | Gartow, Germany | 53.07 | 11.44 | 70 | 341 |
| HPB | Hohenpeissenberg, Germany | 47.80 | 11.02 | 934 | 131 |
| HTM | Hyltemossa, Sweden | 56.10 | 13.42 | 115 | 150 |
| IPR | Ispra, Italy | 45.81 | 8.64 | 210 | 100 |
| JFJ | Jungfraujoch, Switerland | 46.55 | 7.99 | 3580 | 5 |
| KIT | Karlsruhe, Germany | 49.09 | 8.42 | 110 | 200 |
| KRE | Kresín u Pacova, Czech Republic | 49.57 | 15.08 | 534 | 250 |
| LIN | Lindenberg, Germany | 52.17 | 14.12 | 73 | 98 |
| LMP | Lampedusa, Italy | 35.52 | 12.63 | 45 | 8 |
| LUT | Lutjewad, Netherlands | 53.40 | 6.35 | 1 | 60 |
| NOR | Norunda, Sweden | 60.09 | 17.48 | 46 | 100 |
| OPE | Observatoire Pérenne de l'Environnement, France | 48.56 | 5.50 | 390 | 120 |
| OXK | Ochsenkopf, Germany | 50.03 | 11.81 | 1022 | 163 |
| PAL | Pallas, Finland | 67.97 | 24.12 | 565 | 12 |
| PUY | Puy de Dôme, France | 45.77 | 2.97 | 1465 | 10 |
| SAC | Saclay, France | 48.72 | 2.14 | 160 | 100 |
| SMR | Hyytiälä, Finland | 61.85 | 24.29 | 181 | 125 |
| STE | Steinkimmen, Germany | 53.04 | 8.46 | 29 | 252 |
| SVB | Svartberget, Sweden | 64.26 | 19.78 | 269 | 150 |
| TOH | Torfhaus, Germany | 51.81 | 10.54 | 801 | 147 |
| TRN | Trainou, France | 47.96 | 2.11 | 131 | 180 |
| UTO | Utö, Finland | 59.78 | 21.37 | 8 | 57 |
| ZEP | Zeppelin, Swalbard, Norway | 78.91 | 11.89 | 474 | 15 |


**Table 2**. The prior fluxes and $\delta^{13}C$ value used for each source where the total and mean $\delta^{13}C$ values are given for the European domain.

| Source | Total | Dataset/Reference | Mean $\delta^{13}C$ | Reference |
|---|---|---|---|---|
| Agriculture and waste | 24.5 | EDGAR-v5 | -63.0‰ | (Schwietzke et al., 2016) |
| Fossil fuel | 13.5 | EDGAR-v5 | -44.5‰ | (Schwietzke et al., 2016) |
| Wetlands and termites | 5.0 | LPX-Bern | -69.0‰ | (Fisher et al., 2017) |
| Soil sink | -1.0 | LPX-Bern | -22.0‰ | (Reeburgh et al., 1997) |
| Biomass burning | 0.13 | GFED-v4.1s | -22.0‰ | (Schwietzke et al., 2016) |
| Ocean | 0.17 | Weber et al. 2019 | -48.6‰ | (Yu, 2015) |




**Figure 1**. Map of the total mean SRR for optimal sites for scenarios 1 and 2. Since only the EU27+3 emissions are constrained the SRR is also only shown for EU27+3. The locations of the optimal sites are indicated by the white points (i.e., IPR, SAC, KIT, LIN and KRE).

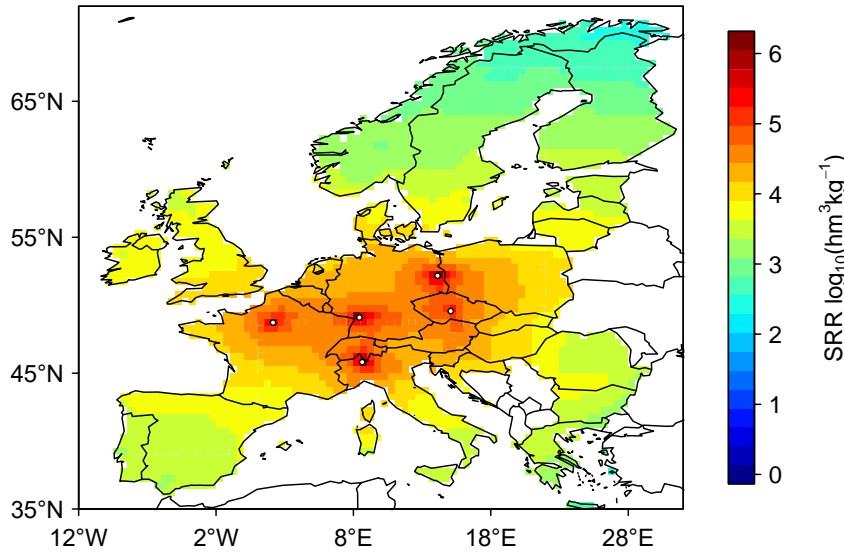

**Figure 2**. Map of annual mean $CH_4$ emissions (units of kg $m^{-2}$ $y^{-1}$) plotted with a logarithmic (base of 2) colour scale.

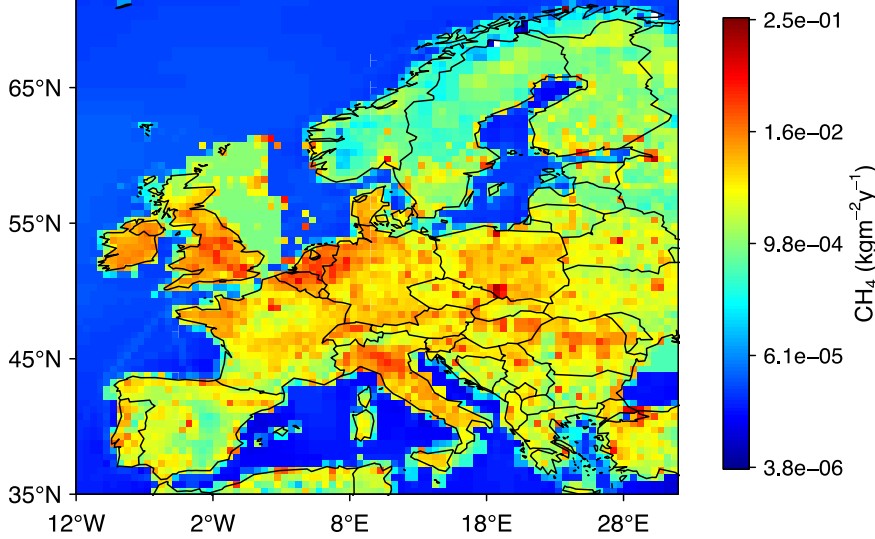




**Figure 3**: Maps showing the uncertainty at each site from the projection of flux uncertainty into the observation space a) $CO_2$ mixing ratios, and b) $\Delta^{14}CO_2$.

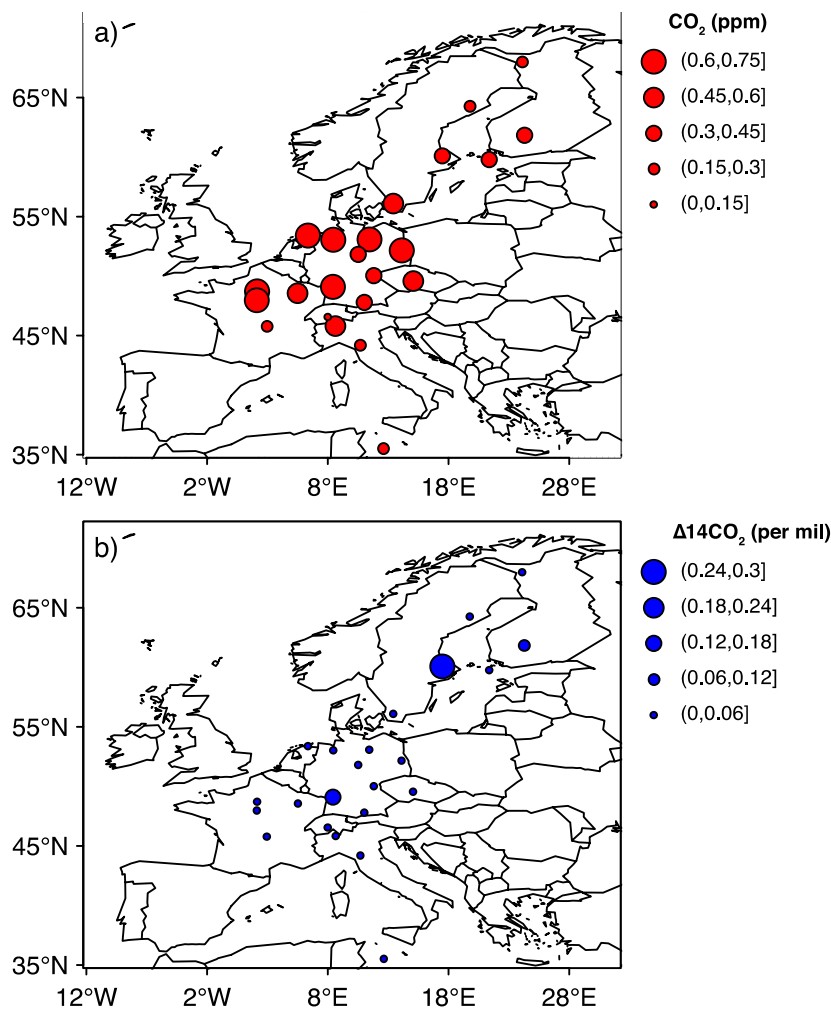




**Figure 4**. Map of the total mean SRR for optimal sites for monitoring fossil fuel $CO_2$ emissions with monthly resolution and including existing information from $CO_2$ mixing ratios. The locations of the optimal sites are indicated by the white points (i.e., SAC, KIT, LUT, KRE, STE, LIN, GAT, IPR, TRN and TOH).

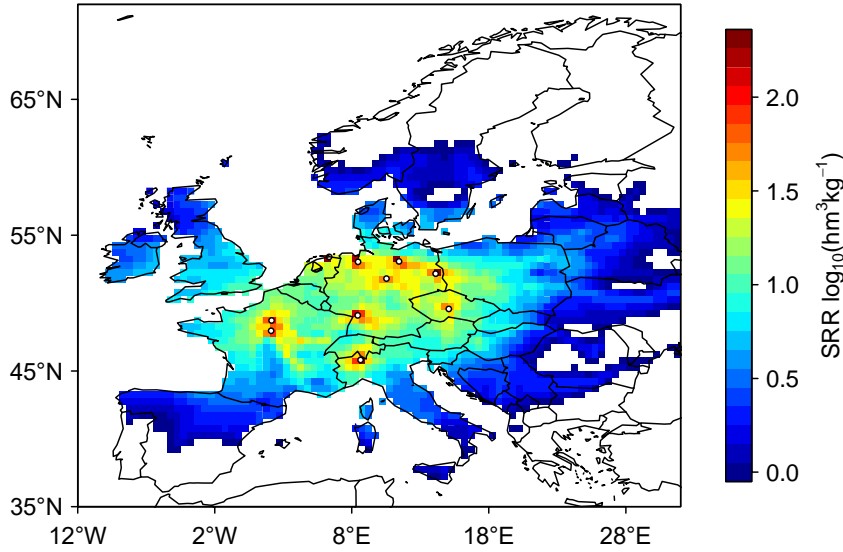

**Figure 5**. Map of annual mean fossil fuel $CO_2$ emissions (units of kg m$^{-2}$ y$^{-1}$) plotted with a logarithmic (base of 2) colour scale.

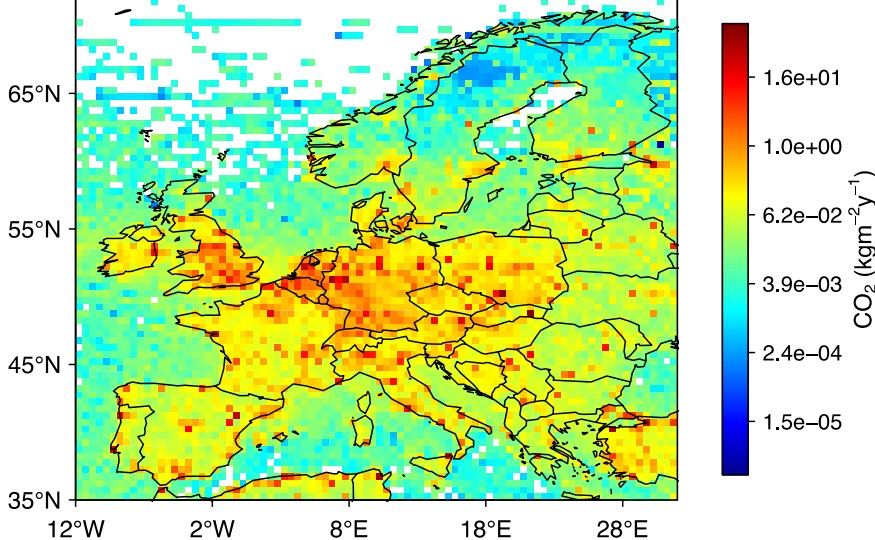