# Peer review of "A Flexible Algorithm for Network Design Based on Information Theory"

_EGUsphere, 2022_

## Referee Comment (RC1)

This paper presents a metric for the evaluation of observing networks. Previous studies have used some scalar function of the posterior covariance. Most commonly this is also the uncertainty in some linear functional of the posterior estimate (such as the total flux over a given region) but metrics like the trace of the posterior covariance are also used. The metric proposed by this paper is the information content defined as the difference in entropy between prior and posterior. Under the common linear Gaussian assumption this turns out to be closely related to the log of the generalised variance or determinant of the posterior covariance.

The authors develop the necessary mathematics for their metric then present two examples, adding isotope measurements for methane or radiocarbon measurements for estimating fossil fuel $CO_2$ emissions. The mathematics is presented well and the examples are clear and pertinent. Furthermore the paper clearly lies within scope for the journal.

My concern with the paper is its lack of evaluation of the metric itself. There is a comparison with the independence metric but not with the covariance metrics. It is not clear to me that the information theory metric does the same job as the covariance metric or a better or worse job. There are two related problems:

First the generalised variance is one metric and probably not a very flexible one. It must include all potential sources. Normally this is not what we want. We have some target quantity like national emissions for which we are designing the network. Normally our target quantity will be a subset of pixels (e.g. pixels in one country). I'm not sure we can easily calculate the determinant of a submatrix of the Hessian.

Next there is the choice of uncertainty quantity to minimise. Rayner et al. (1996) pointed out that the preferred network depended on details of this quantity, such as the total ocean flux vs the average uncertainty for each ocean basin. The determinant is the volume of the hyperellipse described by the posterior covariance. It might be a good general choice but is likely to obscure these differences.

Finally I think the computational advantages of the new metric need a bit more justification. The authors claim that the covariance metric requires the inversion of a large matrix. Depending on the uncertainty metric we choose to minimise this might not be true. In general our target quantity is a linear functional of the posterior sources. Examples include the sum over some subregion and average over time. From what I learned to call Riesz's Representation Theorem (though there seem to be several of these) for any

linear functional $f$ on $R^n$ there is a vector $\mathbf{v}$ such that $f(\mathbf{x}) = \mathbf{v} \cdot \mathbf{x}$ for all $\mathbf{x} \in R^n$. Thus for any target quantity $t$ we can find some vector $\mathbf{v}_t$ such that $t^a = \mathbf{v}_t \cdot \mathbf{x}^a$. The superscript a refers to the analysis or posterior. An example $\mathbf{v}_t$ contains 1 for every pixel in a region and 0 otherwise. This will sum over the region of interest. By the Jacobian law of probabilities the uncertainty in $t$ is given by $\mathbf{v}_t^t \cdot \mathbf{A} \cdot \mathbf{v}_t$ where the superscript T denotes transpose and $\mathbf{A}$ is the posterior or analysis covariance. $\mathbf{A}$ is the inverse of the Hessian $\mathbf{G}$ so we need to calculate $\mathbf{v}_t^T \cdot \mathbf{G}^{-1} \cdot \mathbf{V}_t$. I believe this calculation can be efficiently accomplished by the Cholesky decomposition of $\mathbf{G}$. If we write $\mathbf{G} = \mathbf{L}\mathbf{L}^T$ (Cholesky decomposition) then I believe $\mathbf{G}^{-1} = \mathbf{L}^{-1,T}\mathbf{L}^{-1}$. Substituting this we see $t = \mathbf{y}^T \cdot \mathbf{y}$ where $\mathbf{y} = \mathbf{L}^{-1} \cdot \mathbf{v}_t$. Thus I think the target uncertainty can be performed with a Cholesky decomposition, a matrix-vector product and a dot-product. This may even be less costly than the determinant via the Cholesky decomposition.

I may just as easily be wrong here but think the comparison of the cost and generality of the new metric cf the existing uncertainty metric does need more consideration than it gets here.

I only have two specific comments on the paper:

**L45** When citing early literature it is probably fair to cite the paper that gave rise to the field, Hardt and Scherbaum (1994).

**L65** Summing over the submatrix does indeed account for the covariance of uncertainty but that isn't it's most important property. This is that it calculates the uncertainty on the summed regional flux rather than the individual pixels.

**References**

Hardt, M. and Scherbaum, F.: The Design of Optimum Networks for Aftershock Recordings, Geophys. J. Int., 117, 716–726, 1994.

Rayner, P. J., Enting, I. G., and Trudinger, C. M.: Optimizing the $CO_2$ Observing Network for Constraining Sources and Sinks, Tellus, 48B, 433–444, 1996.

---

## Author Response (AR1)

Responses to Reviewer Comments

**Reviewer 1**

*This paper presents a metric for the evaluation of observing networks. Previous studies have used some scalar function of the posterior covariance. Most commonly this is also the uncertainty in some linear functional of the posterior estimate (such as the total flux over a given region) but metrics like the trace of the posterior covariance are also used. The metric proposed by this paper is the information content defined as the difference in entropy between prior and posterior. Under the common linear Gaussian assumption this turns out to be closely related to the log of the generalised variance or determinant of the posterior covariance. The authors develop the necessary mathematics for their metric then present two examples, adding isotope measurements for methane or radiocarbon measurements for estimating fossil fuel CO2 emissions. The mathematics is presented well and the examples are clear and pertinent. Furthermore the paper clearly lies within scope for the journal.*

*My concern with the paper is its lack of evaluation of the metric itself. There is a comparison with the independence metric but not with the covariance metrics. It is not clear to me that the information theory metric does the same job as the covariance metric or a better or worse job. There are two related problems:*

*First the generalised variance is one metric and probably not a very flexible one. It must include all potential sources. Normally this is not what we want. We have some target quantity like national emissions for which we are designing the network. Normally our target quantity will be a subset of pixels (e.g. pixels in one country). I'm not sure we can easily calculate the determinant of a submatrix of the Hessian.*

**Response:**
We thank the reviewer for his well-thought through comments, especially about the metric we use. Concerning the flexibility of our metric (i.e., the information content, calculated as the log of the determinant of the ratio of prior to posterior uncertainty), yes it can be used to target specific flux types or regions, such as national emissions. In this case, the Hessian matrix is calculated specifically for the region/flux of interest. In the manuscript, we also give examples of this (see e.g. lines 204 to 221). Here we describe how in our example the metric can be set to target only emissions in EU27+3 countries (although the domain is larger) or set to target only e.g. anthropogenic emissions. Furthermore, we describe how the contribution to changes in the mixing ratios from fluxes that are not the target should be accounted for (see Eq. 21). As far as we are aware, our manuscript is the first to formally describe how the influence of non-targeted fluxes on the observations can be accounted for.

*Next there is the choice of uncertainty quantity to minimise. Rayner (1996) pointed out that the preferred network depended on details of this quantity, such as the total ocean flux vs the average uncertainty for each ocean basin. The determinant is the volume of the hyperellipse described by the posterior covariance. It might be a good general choice but is likely to obscure these differences.*

**Response:**

We chose to the information content metric because it describes the change in probability space from before an observation is made (prior probability) compared to after an observation is made (posterior probability) and thus is more closely linked to the observations themselves than to the choice of metric (e.g. minimum of the mean posterior uncertainty of the target fluxes). However, we agree with the reviewer that it would be interesting to include a comparison to a more standard metric, e.g. minimum of the mean posterior uncertainty.

We now include this comparison for the CH4 example in the Discussion section. Using the posterior uncertainty metric, the selected network is quite different to that using the information content (only one site is common between them). Therefore, to test the performance of each network, we ran two pseudo-data inversions each with the same prior fluxes. The performance was assessed using the ratio of the distance of the posterior from the true fluxes versus the distance of the prior from the true fluxes. We found that the inversion using the network based on information content performed slightly better (i.e., the posterior fluxes were closer to the true fluxes) than that based on the posterior uncertainty.

*Finally I think the computational advantages of the new metric need a bit more justification. The authors claim that the covariance metric requires the inversion of a large matrix. Depending on the uncertainty metric we choose to minimise this might not be true. In general our target quantity is a linear functional of the posterior sources. Examples include the sum over some subregion and average over time. From what I learned to call Riesz's Representation Theorem (though there seem to be several of these) for any linear functional f on $R^n$ there is a vector v such that:*

*$f(x) = vx$ for all x in $R^n$*

*Thus for any target quantity t we can find some vector $v_t$ such that*

*$t^a = v_t x^a$*

*The superscript "a" refers to the analysis or posterior. An example $v_t$ contains "1" for every pixel in a region and "0" otherwise. This will sum over the region of interest. By the Jacobian law of probabilities the uncertainty in t is given by*

*$v_t^T A v_t$*

*where A is the posterior or analysis covariance. A is the inverse of the Hessian, G, so we need to calculate:*

*$v_t^T G^{-1} v_t$*

*I believe this calculation can be efficiently accomplished by the Cholesky decomposition of G if we write (Cholesky decomposition):*

$G = LL^T$

*then I believe:*

$G^{-1} = L^{-1T}L^{-1}$

*Substituting this we see:*

$t = y^T y$

*where*

$y = L^{-1}v_t$

*Thus I think the target uncertainty can be performed with a Cholesky decomposition, a matrix-vector product and a dot-product. This may even be less costly than the determinant via the Cholesky decomposition. I may just as easily be wrong here but think the comparison of the cost and generality of the new metric compared to the existing uncertainty metric does need more consideration than it gets here.*

**Response:**

We have included an analysis of the algorithmic complexity and efficiency of using the information content as the metric at the end of the Discussion section and compare it with the posterior uncertainty metric.

Obtaining the posterior covariance is one of the specific cases that cannot be done without inverting matrices. Even if these are relatively well behaved the cost of the operation (and the theoretical complexity) is of the order of $O(n^3)$. The matrix multiplication M(n) of two n x n matrices is also of asymptotic complexity $O(n^3)$. It is true that faster algorithms exist in theory starting with the celebrated Strassen (1969) that is $O(n^{log2(7)})$; but even for matrix larger that 1000x1000 the time reduction can be less than 10% with respect to the naive algorithm due to hardware optimization and acceleration. Besides the practical considerations related to a particular software/hardware implementation it has been established that the algorithmically complexity and hence the computational cost of the calculation of the determinant is the same as that of matrix multiplication (Strassen, 1969; Aho et al., 1974). Therefore, in comparing the two approaches we can abstract the particularities of the algorithm used for matrix multiplication and of its hardware implementation and simplify the analysis by comparing the number of matrix multiplications ($O(n^3)$ or M(n)), matrix inversions ($O(n^3)$ or M(n)), LU ($2O(n^3)$ /3) or Cholesky ($O(n^3)$ /3) decompositions and determinant calculations ($O(n^3)$ or M(n)).

In the case of synthesis inversion, obtaining the posterior covariance of the maximum a posteriori estimate requires two inversions, two matrix multiplications, one addition and one additional inversion (e.g. Tarantola (2005)). Subsequently for obtaining the covariance metric a Cholesky decomposition, a matrix-vector product and a dot-product will be applied as suggested above.

The cost of the covariance metric algorithm can therefore be analised as follows:

1. The calculation of the Hessian $\mathbf{H^T R^{-1} H + B^{-1}}$ requires
1.1 one inversion of B ~ $O(n^3)$ /3
1.2 one inversion of R ~ $O(k^3)$ /3
1.3 product of three matrices requires: $O(n\,k^2 + n^2\,k)$ ~ $O(n^2)$ if k<<n
This yields G in: $O(n^3)$ /3 + $O(k^3)$ /3 + $O(n\,k^2 + n^2\,k)$

2. The Cholesky decomposition of G to obtain L requires ~ $O(n^3)$ /3
The subsequent matrix-vector product and a dot-product are of strictly lower order than 3

Therefore, the general algorithm suggested above is of complexity:
$2O(n^3)$ /3 + $O(k^3)$ /3.
The terms in $O(n^2)$ and lower order can be neglected.

The metric of information content requires the calculation of the Cholesky decomposition of the matrix B and the matrix $H^T R^{-1} H + B^{-1}$ and the trace of their logarithms (both linear).
The calculation of $B^{-1}$ from B requires: $O(n^3)/3$ operations
The calculation of $R^{-1}$ from R requires: $O(k^3)/3$ operations
The calculation of: $\mathbf{H^T R^{-1} H}$ requires: $O(n\,k^2 + n^2\,k)$ ~ $O(n^2)$ if k<<n.

Then remains the calculation of the determinant of the Hessian matrix:
$$|\mathbf{H^T R^{-1} H + B^{-1}}|$$
Which given that it is symmetric and positive definite takes $O(n^3)/3$ operations. The subsequent logarithm and the trace operations and are linear i.e. $O(n)$.

Therefore, the total complexity yields:
$O(n^3)$ /3 + $O(k^3)/3$ + $O(n^2)$ + $O(n^3)$ /3 + $O(n)$ = $2O(n^3)/3$ + $O(k^3)$ /3

As the computational complexities of both procedures are comparable, we can state that the information content procedure is not worse than the posterior uncertainty procedure. We would like to underline that in the submitted manuscript we pointed out that the calculation done in this way is more computationally efficient than applying naively the definition of information content to both the posterior and the prior and then calculating separately the determinant for each of those, and to the best knowledge of the authors this is a novel remark of the present work.

We could remark that information content procedure is intended to be applied to a reduced observation space. In contrast, the general procedure proposed above is intended to be applied to the full space of observations first and then form this generic point the space is reduced with the linear form $v_t$. Any target quantity as a linear function of the posterior sources can be calculated, but the generality of the approach brings the cost of calculating the inverse of the full matrix B. In case of a reduced B the information content metric could be less costly than a generic uncertainty metric applicable to any linear combination of sources. Aggregation of observations would conduce to equivalent reductions in the dimension of R for both algorithms and therefore are not analysed here.

Strassen, V.: "Gaussian Elimination is not Optimal". Numer. Math. **13** (4): 354–356, 1969. doi:10.1007/BF02165411. S2CID 121656251.

Aho, A.V., Hopcroft, J.E. and Ullman, J.D.: "The Design and Analysis of Computer Algorithms". Addison-Wesley, 1974.

*I only have two specific comments on the paper:*

*1) L45: When citing early literature it is probably fair to cite the paper that gave rise to the field, Hardt et al. 1994*

**Response:**
We presume the reviewer is referring to this paper:
Hardt, M. and Scherbaum, F.: The design of optimum networks for aftershock recordings, Geophysical Journal International, 117, (3), 1994. https://doi.org/10.1111/j.1365-246X.1994.tb02464.x
In this case, we now refer to this paper in the introduction (L44).

*2) L64: Summing over the submatrix does indeed account for the covariance of uncertainty but that isn't it's most important property. This is that it calculates the uncertainty on the summed regional flux rather than the individual pixels.*

**Response:**
We thank the reviewer for pointing this out. We now also mention at L63 that summing over a submatrix of the posterior uncertainty provides the regional uncertainty.

**Reviewer 2:**

*Overview:*
*The manuscript "A Flexible Algorithm for Network Design Based on Information Theory" by Thompson and Pisso describes the development of a novel method for optimising the distribution of a measurement network, with the aim of maximising the information content provided by these measurements for a flux inversion. Previous methods, usually based on quantifying the posterior uncertainty of the inversion, were computationally expensive but the metric presented here should be more efficient. The new method is applied to improving the current European measurement network for CH4 and CO2 through inclusion of isotopic measurements at a subset of locations. The paper is well-written and presented, with thorough explanation of the methodology and clear figures. The new method appears to provide a justifiable technique for network design.*

*My only significant comment is that the discussion of the results of the new method in the context of previous methods is very brief. The results are compared to those using the clustering-based selection discussed earlier in the text, which is based on discounting sites with similar observed signals. However, there is no comparison of the merits of the new method compared to those based on posterior uncertainty. Whilst, for computational reasons, I understand that the authors might not want to explicitly perform such an analysis*

*for direct comparison, I do think that there needs to be some further discussion of the potential differences, advantages and disadvantages of the new method compared to the the full range of alternative methods. If the last sections are expanded to include such discussion, I am happy to recommend this manuscript for publication in this journal.*

**Response:**

We thank the reviewer for his/her thoughtful comments. We have expanded the Discussion section to include a comparison of the information content metric with the posterior uncertainty metric for the example of CH4 fluxes. This was also included in response to a comment by reviewer 1.

***Minor/technical comments:***

*line 30: brackets around reference year*

*line 70: slightly unclear. heterogeneity in terms of flux?*

*figure 1: It would be good to also mark the locations of the sites that were not selected by the algorithm in Figure 1. I appreciate that they are shown in a later figure, but it is easiest for the reader, and would aid comprehension of Fig. 1, if they are noted earlier than later.*

*figure 5: Is it possible to say anything in the main text concerning why the two sites located very close to each other in France might have been selected using this method?*

**Responses:**

L30: done

L70: Actually it is the heterogeneity of the atmosphere, we now specify this.

Figure 1: We now include both locations of unselected and selected sites in Fig. 1 for CH4 and Fig. 4, the equivalent figure for CO2.

Figure 5: The two French sites are SAC (Saclay, located just south of Paris) and TRN (Trainou, located approximately 95 km south of SAC. The two sites have slightly different footprints with SAC sampling more the Paris region and TRN sampling more to the south and east. For this reason, and considering the fairly large emissions from these regions of France, both sites are selected. However, the choice of TRN is somewhat dependent on the set-up, e.g. if the fluxes are only resolved annually, then TRN is no longer selected but rather HPB, which also indicates that the information from selecting this site is similar to that of some other candidate sites (i.e. HPB). We have now included this explanation in the last paragraph of section 3.2.